# Pyruvate:ferredoxin oxidoreductase and low abundant ferredoxins support aerobic photomixotrophic growth in cyanobacteria

Yingying Wang[1], Xi Chen[1], Katharina Spengler[1], Karoline Terberger[1], Marko Boehm[1,2], Jens Appel[1,2], Thomas Barske[3], Stefan Timm[3], Natalia Battchikova[4], Martin Hagemann[3], Kirstin Gutekunst[1,2]*

[1]Department of Biology, Botanical Institute, Christian-Albrechts-University, Kiel, Germany; [2]Department of Molecular Plant Physiology, Bioenergetics in Photoautotrophs, University of Kassel, Kassel, Germany; [3]Plant Physiology Department, University of Rostock, Rostock, Germany; [4]Department of Biochemistry, Molecular Plant Biology, University of Turku, Turku, Finland

**Abstract** The decarboxylation of pyruvate is a central reaction in the carbon metabolism of all organisms. It is catalyzed by the pyruvate:ferredoxin oxidoreductase (PFOR) and the pyruvate dehydrogenase (PDH) complex. Whereas PFOR reduces ferredoxin, the PDH complex utilizes NAD+. Anaerobes rely on PFOR, which was replaced during evolution by the PDH complex found in aerobes. Cyanobacteria possess both enzyme systems. Our data challenge the view that PFOR is exclusively utilized for fermentation. Instead, we show, that the cyanobacterial PFOR is stable in the presence of oxygen in vitro and is required for optimal photomixotrophic growth under aerobic and highly reducing conditions while the PDH complex is inactivated. We found that cells rely on a general shift from utilizing NAD(H)- to ferredoxin-dependent enzymes under these conditions. The utilization of ferredoxins instead of NAD(H) saves a greater share of the Gibbs-free energy, instead of wasting it as heat. This obviously simultaneously decelerates metabolic reactions as they operate closer to their thermodynamic equilibrium. It is common thought that during evolution, ferredoxins were replaced by NAD(P)H due to their higher stability in an oxidizing atmosphere. However, the utilization of NAD(P)H could also have been favored due to a higher competitiveness because of an accelerated metabolism.

*For correspondence:
kirstin.gutekunst@uni-kassel.de

Competing interest: The authors declare that no competing interests exist.

## Editor's evaluation

In this study, the authors detail evidence supporting a role of pyruvate:ferredoxin oxidoreductase under oxygenic conditions. The works provides explanations for why "anaerobic enzymes" can be present and advantageous under aerobic conditions.

## Introduction

### FeS clusters, pyruvate:ferredoxin oxidoreductase, and ferredoxins

Life evolved under anaerobic conditions in an environment that was reducing and replete with iron and sulfur. Later on, hydrogen escape to space irreversibly oxidized Earth (*Zahnle et al., 2013*; *Catling et al., 2001*). Prebiotic redox reactions that took place on the surfaces of FeS minerals, are at present mimicked by catalytic FeS clusters in a plethora of enzymes and redox carriers (*Russell*

*and Martin, 2004*; *Wächtershäuser, 1990*). One example are ferredoxins, that are small, soluble proteins containing 4Fe4S, 3Fe4S, or 2Fe2S clusters and shuttle electrons between redox reactions. They display a wide range of redox potentials between −240 to −680 mV and are involved in a variety of metabolic pathways (*Liu et al., 2014*). Ferredoxins are among the earliest proteins on Earth and are accordingly present in all three kingdoms of life (*Kim et al., 2012*). FeS enzymes are especially widespread in anaerobes (*Müller et al., 2012*).

The advent of oxygenic photosynthesis necessitated adaptations, as especially 4Fe4S clusters are oxidized and degraded to 3Fe4S in the presence of oxygen resulting in inactivated enzymes (*Müller et al., 2012*; *Imlay, 2006*; *Jagannathan et al., 2012*). In aerobes, FeS enzymes are commonly replaced by FeS cluster-free isoenzymes or alternative metabolic strategies (*Imlay, 2006*). One well-known example is the replacement of the FeS cluster containing pyruvate:ferredoxin oxidoreductase (PFOR), which catalyzes the decarboxylation of pyruvate during fermentation in anaerobes, by the pyruvate dehydrogenase (PDH) complex for respiration in aerobes (*Müller et al., 2012*; *Gould et al., 2019*). Both enzymes catalyze the same reaction, however, PFOR uses ferredoxin as redox partner and the PDH complex reduces $NAD^+$. PFORs are old enzymes from an evolutionary point of view. They are widespread in autotrophic and heterotrophic bacteria, in archaea, amitochondriate eukaryotic protists, hydrogenosomes as well as in cyanobacteria and algae (*Müller et al., 2012*). Depending on organism, metabolism and conditions, PFOR can be involved in the oxidation of pyruvate for heter-otrophy or alternatively catalyze the reverse reaction by fixing $CO_2$ and forming pyruvate from acetyl CoA for an autotrophic lifestyle (*Witt et al., 2019*; *Mall et al., 2018*; *Evans et al., 1966*). The enzyme might have played a central role for the evolution of both autotrophic and heterotrophic processes from the very beginning (*Gutekunst, 2018*). PFOR indeed participates as $CO_2$ fixing enzyme in four out of seven currently known and most ancient autotrophic pathways (reverse tricarboxylic acid [rTCA] cycle, reversed oxidative tricarboxylic acid [roTCA] cycle, reductive acetyl-CoA pathway, and dicar-boxylate/hydroxybutyrate [DC/HB] cycle) (*Mall et al., 2018*; *Fuchs, 2011*). PFORs contain one to three 4Fe4S clusters and in general get inactivated readily by oxygen upon purification. So far, there are only three reported exceptions to this rule: the PFORs of *Halobacterium halobium*, *Desulvovibrio africanus*, and *Sulfolobus acidocaldarius* (*Witt et al., 2019*; *Pieulle et al., 1995*; *Pieulle et al., 1997*; *Vita et al., 2008*; *Kerscher and Oesterhelt, 1981*). Even though all three enzymes are stable upon purification in the presence of oxygen, anaerobic conditions are required for in vitro maintenance of enzyme activities with the PFORs of *D. africanus* and *S. acidocaldarius*. The enzyme of *H. halobium* is the only PFOR reported so far, which is active under aerobic conditions in vitro (*Kerscher and Oesterhelt, 1981*; *Kerscher and Oesterhelt, 1977*). In vivo studies on these PFORs under aerobic conditions are missing. Recently it was shown in *Escherichia coli*, that PFOR plays an important role in aerobic cultures of a mutant in which glucose-6P dehydrogenase (ZWF) was downregulated (*Li et al., 2021*). PFOR is probably involved in redox control during stationary phase in this mutant (*Li et al., 2021*). This finding is highly surprising, as PFOR activity in *E. coli* crude extracts is only detectable under anaerobic conditions in vitro (*Nakayama et al., 2013*). There are several reports on the aerobic expression of enzymes that are assigned to anaerobic metabolism in prokaryotes and eukaryotes and therefore challenge the simplistic distinction between aerobic versus anaerobic enzymes (*Gould et al., 2019*; *Schmitz et al., 2001*; *Gutekunst et al., 2005*). Their physiological significance and regu-lation are only partly understood.

Ferredoxins that contain 4Fe4S clusters are likewise vulnerable to oxidative degradation. In the evolution from anoxygenic to oxygenic photosynthesis, the soluble 4Fe4S ferredoxin, which transfers electrons from FeS-type photosystems PSI (photosystem I) to other enzymes in anoxygenic photosyn-thesis was replaced by an oxygen-tolerant 2Fe2S ferredoxin (*Jagannathan et al., 2012*). In addition, NAD(P)H has gained importance as alternative, oxygen-insensitive reducing agent in aerobes and thereby complemented or replaced oxygensensitive ferredoxins, that are useful for anaerobes (*Gould et al., 2019*).

## The PDH complex

The PDH complex, which utilizes $NAD^+$ is composed of the three subunits: PDH (E1), dihydrolipoyl transacetylase (E2), and dihydrolipoyl dehydrogenase (E3). It catalyzes the irreversible decarboxyl-ation of pyruvate. The PDH complex is active under oxic conditions but gets inactivated under anaer-obic conditions in both prokaryotes and eukaryotes, albeit via distinct mechanisms. In the absence

of oxygen NADH/NAD$^+$ ratios rise as respiration no longer oxidizes the NADH coming from the PDH complex and the subsequent reactions of the TCA cycle. In prokaryotes, as for example *E. coli*, NADH interacts with the dihydrolipoyl dehydrogenase (E3) subunit and thereby inhibits the PDH complex (***Kim et al., 2008***; ***Sun et al., 2012***). In eukaryotes, the PDH complex gets inactivated at high NADH/NAD$^+$ ratios via phosphorylation of highly conserved serine residues in the PDH (E1) subunit (***Kolobova et al., 2001***).

*Synechocystis* sp. PCC 6803 is a cyanobacterium that performs oxygenic photosynthesis and lives photoautotrophically by fixing $CO_2$ via the Calvin–Benson–Bassham (CBB) cycle. In the presence of external carbohydrates these are metabolized additionally, resulting in a photomixotrophic lifestyle. In darkness *Synechocystis* switches to a heterotrophic or under anaerobic conditions to a fermentative lifestyle. As in many cyanobacteria, pyruvate can be either decarboxylated via PFOR or alternatively via the PDH complex in *Synechocystis*. PFOR is assumed to be involved in fermentation under anoxic conditions and the PDH complex in aerobic respiration. The observation that *pfor* is transcribed under photoautotrophic conditions in the presence of oxygen in the cyanobacteria *Synechococcus* sp. PCC 7942 and *Synechocystis* was therefore surprising but is well in line with the observation that other enzymes assigned to anaerobic metabolism in eukaryotes are expressed in the presence of oxygen as well (***Gould et al., 2019***; ***Schmitz et al., 2001***). *Synechoystis* possesses a network of up to 11 ferredoxins containing 2Fe2S, 3Fe4S, and 4Fe4S clusters (***Cassier-Chauvat and Chauvat, 2014***; ***Artz et al., 2020***). The 2Fe2S ferredoxin 1 (Ssl0020) is essential and by far the most abundant ferredoxin in *Synechocystis* and is the principal acceptor of photosynthetic electrons at PSI (***Bottin and Lagoutte, 1992***). Structures, redox potentials and distinct functions have been resolved for some of the alternative low abundant ferredoxins, however, the metabolic significance of the complete network is still far from being understood (***Cassier-Chauvat and Chauvat, 2014***; ***Artz et al., 2020***; ***Gutekunst et al., 2014***; ***Schorsch et al., 2018***; ***Motomura et al., 2019***; ***Mustila et al., 2014***).

In this study, we show that PFOR and low abundant ferredoxins are required for optimal photomixotrophic growth under oxic conditions. In line with this we found that the cyanobacterial PFOR is stable in the presence of oxygen in vitro. PFOR and ferredoxins can functionally replace the NAD$^+$-dependent PDH complex, which we found is inactivated at high NADH/NAD$^+$ ratios. Likewise, the ferredoxin-dependent F-GOGAT (glutamine oxoglutarate aminotransferase) is essential for photomixotrophic growth as well and cannot be functionally replaced by the NADH-dependent N-GOGAT. The cells obviously switch in their utilization of isoenzymes and redox pools. However, the key factor for this switch is not oxygen but are the highly reducing conditions within the cells. Our data suggest that the pool of ferredoxins in *Synechocystis* functions as an overflow basin to shuttle electrons, when the NADH/NAD$^+$ pool is highly reduced.

## Results

The roles of PDH complex and PFOR were studied in *Synechocystis* under different growth conditions. PDH could not be deleted from the genome indicating that this enzyme complex is essential, whereas *pfor* was knocked out in a previous study (***Gutekunst et al., 2014***). In line with this, we found that all fully sequenced diazotrophic and nondiazotrophic cyanobacteria with photosystem II (PSII contain genes coding for a PDH complex and that 56% of these cyanobacteria possess a PFOR as well). If we subtract from this group all diazotrophic cyanobacteria that contain a nitrogenase and might therefore utilize PFOR in the process of nitrogen fixation, 130 nondiazotrophic cyanobacteria remain. Within the group of nondiazotrophic cyanobacteria 42% possess a PFOR in addition to the PDH complex (***Figure 1—figure supplement 1***). This clearly shows that the property of holding both a PDH complex and a PFOR in cyanobacteria that live predominantly under oxic conditions is truly widespread. The analysis furthermore confirms our observation, that the PDH complex is preferred over the utilization of PFOR in cyanobacteria.We unexpectedly found that the *Synechocystis* Δ*pfor* deletion mutant was impaired in its photomixotrophic growth under oxic conditions in continuous light. Growth impairment was typically visible starting around days 3–6 of the growth experiment (***Figure 1A*** and 3A). In addition, maintenance in the stationary growth phase was affected in Δ*pfor*. Under photoautotrophic conditions Δ*pfor* grew similar to the WT (***Figure 1A***). The oxygen concentration in the photomixotrophic cultures was close to saturation around 250 µMol $O_2$ throughout the growth experiment (***Figure 1—figure supplement 2***). Studies on the transcription of *pfor* and the alpha subunit of the PDH (E1) of *pdhA* revealed that both genes are transcribed under photomixo- and photoautotrophic

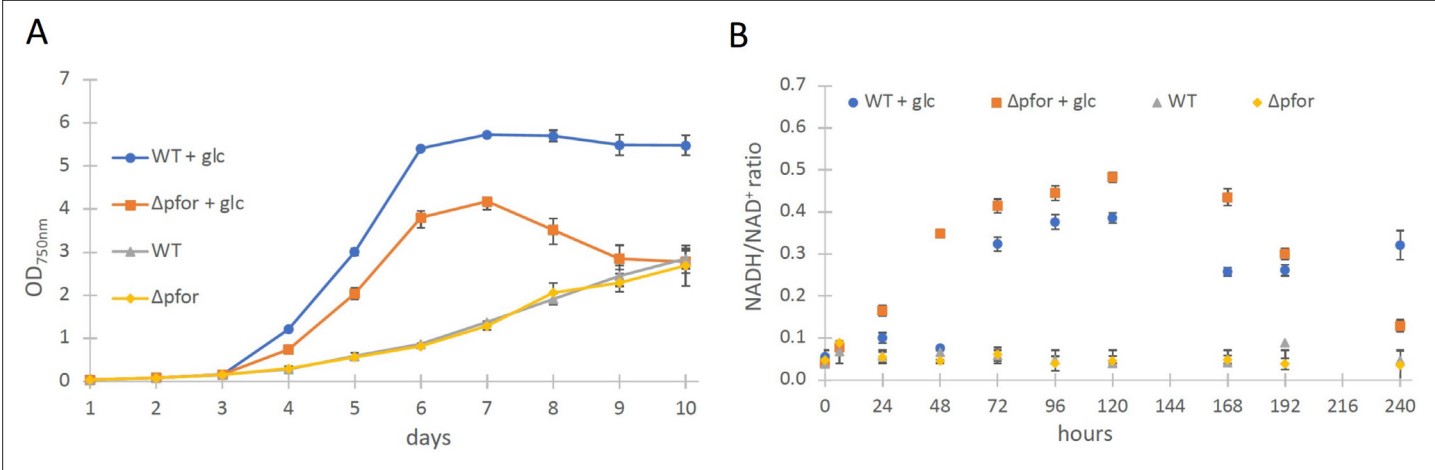

**Figure 1.** Wild type and Δpfor under photoautotrophic and photomixotrophic conditions. (**A**) Growth and (**B**) NADH/NAD+ ratios of wild type (WT) and Δ*pfor* under photoautotrophic and photomixotrophic (+glc) conditions in continuous light. Shown are mean values ± standard deviation (SD) from at least three replicates.

The online version of this article includes the following source data and figure supplement(s) for figure 1:

**Source data 1.** Raw data of growth and NADH/NAD+ ratio of wild type (WT) and Δpfor under photoautotrophic and photomixotrophic conditions.

**Figure supplement 1.** Bioinformatic analyses concerning the distribution of pyruvate dehydrogenase (PDH) complex and pyruvate:ferredoxin oxidoreductase (PFOR) in diazotrophic and nondiazotrophic cyanobacteria.

**Figure supplement 1—source data 1.** Raw data of bioinformatic analysis of the occurrence of pyruvate:ferredoxin oxidoreductase (PFOR) and the pyruvate dehydrogenase (PDH) complex in cyanobacteria.

**Figure supplement 2.** Oxygen concentrations in photomixotrophic cultures of wild type (WT) and *pfor* were close to oxygen saturation throughout the growth experiments.

**Figure supplement 2—source data 1.** Raw data of oxygen concentration in photomixotrophic wild type (WT) and Δpfor cultures.

**Figure supplement 3.** RT-PCR showing that *pfor* and *pdhA* are transcribed under photoautotrophic and photomixotrophic conditions in the wild type.

**Figure supplement 3—source data 1.** Uncropped raw gel of RT-PCR.

**Figure supplement 4.** Redox states of NAD(P)H and ferredoxin and $O_2$ turnover in auto- and mixotrophic cultures.

conditions (*Figure 1—figure supplement 3*). These observations raised two questions: Why is the PDH complex, which catalyzes the same reaction as PFOR, not able to compensate for the loss of PFOR? And how can PFOR, which is assumed to be oxygen sensitive, be of physiological relevance in the presence of oxygen?

The most obvious assumption is that the PDH complex might get inactivated under photomixotrophic conditions. As the PDH complex gets inactivated at high NADH/NAD+ ratios in prokaryotes and eukaryotes (*Kim et al., 2008*; *Sun et al., 2012*; *Kolobova et al., 2001*), we wondered if NADH/NAD+ ratios might be increased under photomixotrophic conditions. Corresponding measurements confirmed this assumption. Whereas NADH/NAD+ ratios were stable under photoautotrophic conditions in WT and Δ*pfor* they raised three to fourfold in the first 5 days of photomixotrophic growth, exactly in that period in which the growth impairment of Δ*pfor* in the presence of glucose was most apparent (*Figure 1B*).

In addition, in vivo NAD(P)H fluorescence measurements and estimates for the reduction level of ferredoxin using a Dual-KLAS/NIR were performed, which show that in addition to the NADH/NAD+ ratio, also the NAD(P)H and ferredoxin pools are more reduced under photomixotrophic conditions in comparison to photoautotrophic conditions (*Figure 1—figure supplement 4*).

For prokaryotes it was shown that the PDH complex is inhibited by a distinct mechanism directly by NADH which binds to the dihydrolipoyl dehydrogenase (E3) subunit of the PDH complex (*Kim et al., 2008*; *Sun et al., 2012*). Therefore, the recombinant dihydrolipoyl dehydrogenase of *Synechocystis* (SynLPD) was tested in an in vitro assay with different NADH concentrations. The enzyme indeed loses activity at higher NADH/NAD+ ratios, whereas NADPH has no effect (*Figure 2A*). The SynLPD activity was completely inhibited by NADH with an estimated $K_i$ of 38.3 µM (*Figure 2A*). Hence, the enzyme

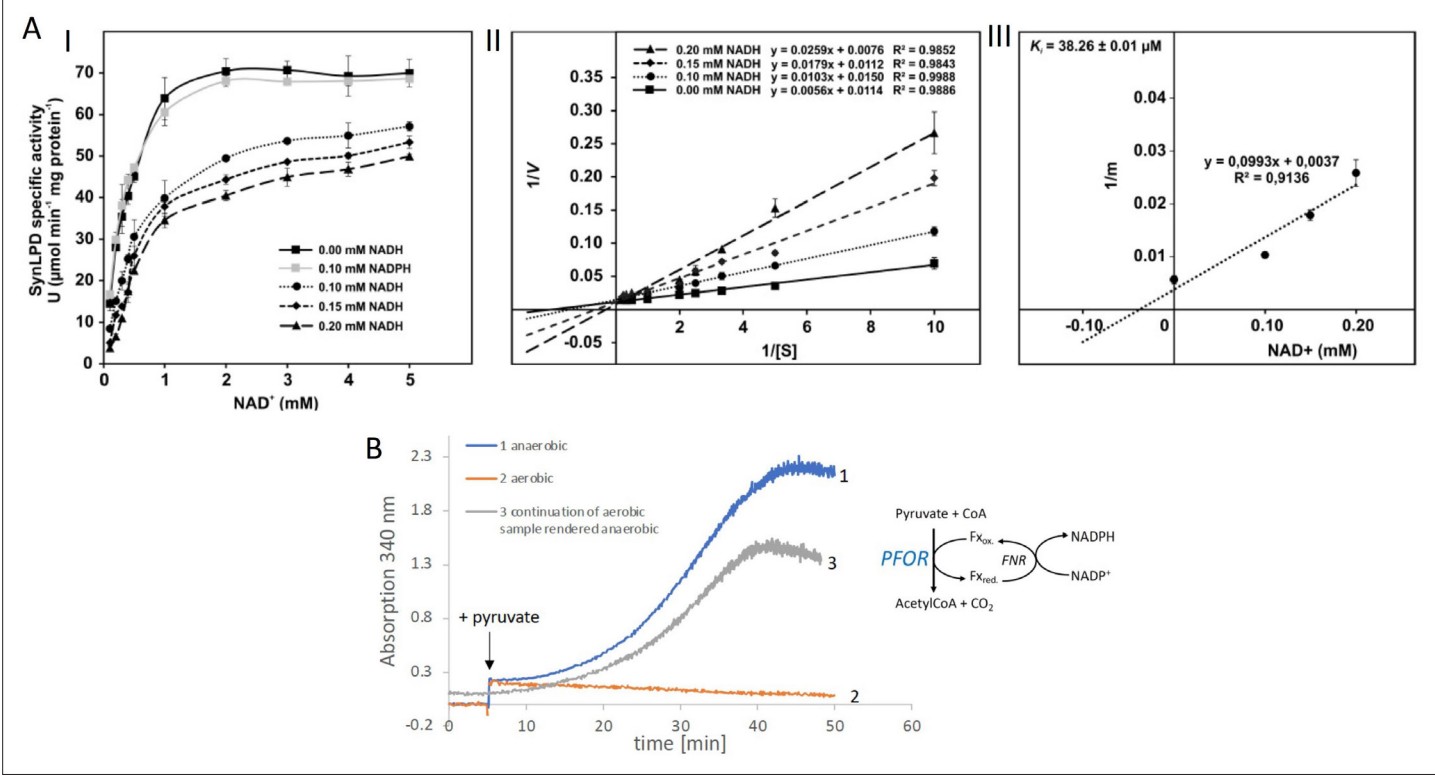

**Figure 2.** Biochemical characterization of the pyruvate dehydrogenase complex and the pyruvate:ferredoxin oxidoreductase. (**A**) Inhibition of the pyruvate dehydrogenase (PDH) complex in *Synechocystis* via inactivation of the dihydrolipoyl dehydrogenase (E3) subunit (SynLPD) by NADH. I: The rate of recombinant SynLPD activity (3 mM DL-dihydrolipoic acid) as a function of $NAD^+$ (0.1, 0.2, 0.3, 0.4, 0.5, 1, 2, 3, 4, and 5 mM) reduction in the presence of the indicated NADH concentrations (0, 0.1, 0.15 and 0.2 mM). NADPH (0.1 mM) was used as a control to demonstrate the specificity of NADH inhibition. Specific enzyme activity is expressed in µmol NADH per $min^{-1}$ mg $protein^{-1}$ at 25°C. II: Lineweaver-Burk plots of enzyme activities at four NADH concentrations. III: The inhibitor constant (Ki) was estimated by linear regression of (I) the slopes of the three Lineweaver-Burk plots at the four NADH concentrations versus (II) the NADH concentration. Shown are mean values ± standard deviation (SD) from at least 3 technical replicates. (**B**) Enzyme activity of pyruvate:ferredoxin oxidoreductase (PFOR) that was purified in the presence of oxygen. PFOR activity was measured in the presence of ferredoxin-NADPH-oxidoreductase (FNR), ferredoxin, and NADP+. The reaction was started by addition of 10 mM pyruvate as indicated by the arrow. Assay 1 (blue line): The assay mixture was kept anaerobic with 40 mM glucose, 40 U glucose oxidase, and 50 U catalase, showing that PFOR, which was purified in the presence of oxygen, is active. Assay 2 (red line): Assay 2 had the same composition as assay 1 but glucose, glucose oxidase and catalase were omitted, showing that anaerobic conditions are required for activity of PFOR in vitro. Assay 3 (gray line): This assay is the continuation of the measurement of assay 2 after addition of glucose, glucose oxidase and catalase. Representative traces of three replicates are shown.

The online version of this article includes the following source data and figure supplement(s) for figure 2:

**Source data 1.** Raw data of enzymatic in vitro test with the dihydrolipoyl dehydrogenase (E3) subunit (SynLPD) of the pyruvate dehydrogenase (PDH) complex and pyruvate:ferredoxin oxidoreductase (PFOR).

**Figure supplement 1.** Sodium dodecyl sulfate–polyacrylamide gel electrophoresis (SDS–PAGE) analysis followed by immunoblotting of *Synechocystis* soluble extracts.

**Figure supplement 2.** Large-scale pyruvate:ferredoxin oxidoreductase (PFOR) purification.

**Figure supplement 2—source data 1.** Uncropped gel of pyruvate:ferredoxin oxidoreductase (PFOR) purification.

**Figure supplement 3.** PCR analysis of pyruvate:ferredoxin oxidoreductase overexpression (pfor:oe) mutant and wild type (WT).

activity dropped to approximately 50% at a NADH/$NAD^+$ ratio of 0.1 (e.g., at 0.2 mM NADH in the presence of 2 mM $NAD^+$). Please note, that much higher NADH/$NAD^+$ ratios (>0.4) were measured in photomixotrophic cells of *Synechocystis* (see *Figure 1B*). This points to an efficient inhibition of PDH activity via the highly decreased function of the E3 subunit (SynLPD). NADH/$NAD^+$ ratios above 0.1 could not be tested in the enzyme assays due to the high background absorption of the added NADH, which prevented SynLPD activity detections.

Taken together these measurements convincingly show that the PDH complex is most likely inhibited under photomixotrophic conditions at high NADH/$NAD^+$ ratios, which provides evidence that

pyruvate oxidation must be performed instead via PFOR and gives an explanation for the importance of PFOR under these conditions.

As the cyanobacterial PFOR is regarded as an oxygen-sensitive enzyme that exclusively supports fermentation under anaerobic conditions, we overexpressed the enzyme and purified it in the presence of oxygen in order to check for its stability under aerobic conditions (*Figure 2—figure supplement 1*, *Figure 2—figure supplement 2*, *Figure 2—figure supplement 3*). Enzymatic tests revealed that PFOR from *Synechocystis* was indeed stable under aerobic conditions in vitro, which means that the enzyme was not degraded and kept its activity but required anoxic conditions for the decarboxylation of pyruvate (*Figure 2B*) as reported for the oxygen stable PFORs of *D. africanus* and *S. acidocaldarius* (*Witt et al., 2019*; *Pieulle et al., 1995*).

In contrast to the PDH complex, PFOR transfers electrons from pyruvate to oxidized ferredoxin. In order to investigate if any of the low abundant ferredoxins (Fx) might be of importance for photomixotrophic growth, respective deletion mutants were generated (*Supplementary file 1a, b*, *Figure 3—figure supplement 1*) and tested for their ability to grow under photoautotrophic and photomixotrophic conditions. To this end, *fx3* (*slr1828*), *fx4* (*slr0150*), *fx6* (*ssl2559*), *fx7* (*sll0662*), and *fx9* (*slr2059*) could be completely deleted from the genome, whereas the deleted alleles of *fx2* (*sll1382*) and *fx5* (*slr0148*) failed to segregate, keeping some wild-type copies. Furthermore, we did not succeed to delete *fx8* (*ssr3184*). Flavodoxin (*isiB; sll0284*), which replaces ferredoxins functionally under Fe limitation was deleted as well. In addition, the double mutants Δ*fx7*Δ*fx9* and Δ*fx9*Δ*isiB* as well as the triple mutant Δ*fx7*Δ*fx8*Δ*fx9* were generated. Photoautotrophic growth of all these ferredoxin deletion mutants was similar to the WT (*Figure 3—figure supplement 2*). However, under photomixotrophic conditions deletion of either *fx3*, *fx9*, or flavodoxin (*isiB*) resulted in a growth behavior that was similar to Δ*pfor* (*Figure 3A*).

These results indicate that there might be a general shift to utilize the ferredoxin pool as soon as the NADH/NAD⁺ pool is over-reduced. Beside the PFOR/PDH complex couple, GOGAT (<u>g</u>lutamine <u>o</u>xo<u>g</u>lutarate <u>a</u>mino<u>t</u>ransferase) as well is present in form of two isoenzymes in *Synechocystis* that either utilizes reduced ferredoxin (F-GOGAT; *sll1499*) or NADH (N-GOGAT; *sll1502*). In line with our assumption that ferredoxin utilization is preferred in over-reduced cells after glucose addition, we hypothesized that F-GOGAT might be required for optimal photomixotrophic growth. Respective deletion mutants were generated (*Supplementary file 1a, b*, *Figure 3—figure supplement 1*) and revealed that neither Δ*f-gogat* nor Δ*n-gogat* were impaired in their growth under photoautotrophic conditions, whereas Δ*f-gogat* displayed a strong growth impairment under photomixotrophic conditions in contrast to Δ*n-gogat* and the WT (*Figure 3B*). These data indicate that cells indeed rely on a general switch from utilizing NAD(H) to utilizing ferredoxins for optimal photomixotrophic growth. It was recently shown that photosynthetic complex I (NDH1) exclusively accepts electrons from reduced ferredoxin instead of NAD(P)H (*Schuller et al., 2019*). Under photomixotrophic conditions photosynthesis operates in parallel to carbon oxidation. In addition to water oxidation at PSII, electrons from glucose oxidation can as well enter the respiratory/photosynthetic electron transport chain and eventually arrive at PSI. Photosynthesis based on PSI thus uses electrons from glucose oxidation that enter the respiratory/photosynthetic electron transport chain and are excited at PSI.

Three entry points exist that can feed electrons from glucose oxidation into the plastoquinone (PQ) pool in the thylakoid membrane: the succinate dehydrogenase, which accepts electrons from the conversion of succinate to fumarate; NDH-2, which accepts electrons from NADH and photosynthetic complex I (NDH-1), which accepts electrons from reduced ferredoxin (see *Figure 4B*). Based on the observed shift from utilizing ferredoxin instead of NAD(P)H, we thus wondered if photosynthetic complex I (NDH-1) might be required for photosynthesis (involving only PSI) under photomixotrophic conditions as an entry point for electrons coming from glucose oxidation. Cells were incubated with DCMU that blocks the electron transfer from PSII to the PQ pool. Thereby, electron transfer from glycogen or glucose oxidation to PSI could be measured based on a recently developed protocol (*Theune et al., 2021*). According to this protocol electrons were counted that flow through PSI via DIRK$_{PSI}$ measurements by the KLAS/NIR instrument. The electron transport at PSI was then measured in the absence and in the presence of glucose. In addition to the WT, several mutants were analyzed with deletions in entry points as well as glucose metabolizing enzymes. The mutant with a deleted photosynthetic complex I (Δ*ndhD1*Δ*ndhD2*) should no longer be able to feed electrons from reduced ferredoxin into the respiratory/photosynthetic electron transport chain, while the hexokinase mutant

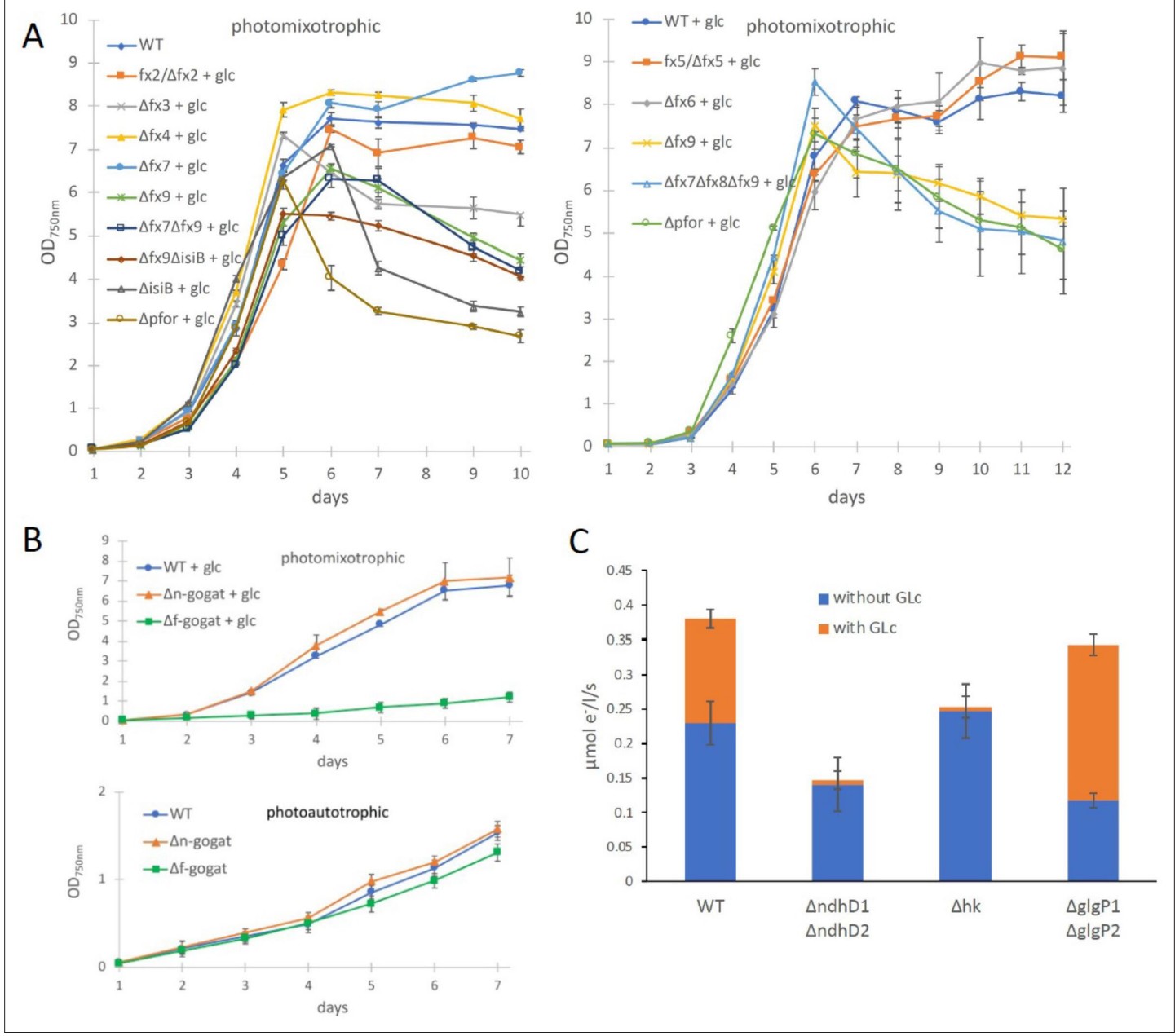

**Figure 3.** Growth and electron transport at PSI in wild type and several deletion mutants in the presence and absence of glucose. (**A**) Photomixotrophic growth of wild type (WT), *Δpfor*, ferredoxin (fx), and flavodoxin (isiB) deletion mutants as indicated. (**B**) Growth of WT, *Δf-gogat*, and *Δn-gogat* under photoautotrophic and photomixotrophic conditions. (**C**) Electron transport with 3-(3,4-dichlorophenyl)-1,1-dimethylurea (DCMU) at photosystem I (PSI) in the absence and presence of glucose in the WT, *ΔndhD1ΔndhD2*, *Δhk*, and *ΔglgP1ΔglgP2*. Shown are mean values ± standard deviation (SD) from at least 3 replicates.

The online version of this article includes the following source data and figure supplement(s) for figure 3:

**Source data 1.** Raw data from growth of wild type (WT), *Δpfor*, Δfx, ΔisiB, *Δf-gogat*, and *Δn-gogat* and electron transport with DCMU at photosystem I (PSI) in the absence and presence of glucose in the WT, *ΔndhD1ΔndhD2*, *Δhk*, and *ΔglgP1**glgP2***.

**Figure supplement 1.** The examination of segregation of mutant strains.

**Figure supplement 1—source data 1.** Uncropped raw gels and blots from the examination of deletion mutants.

**Figure supplement 2.** Photoautotrophic growth of different ferredoxin (fx) and the flavodoxin (isiB) deletion mutant as indicated in comparison to the wild type (WT).

**Figure supplement 2—source data 1.** Raw data from photoautotrophic growth of different ferredoxin (fx) and the flavodoxin (isiB) deletion mutant as indicated in comparison to the wild type (WT).

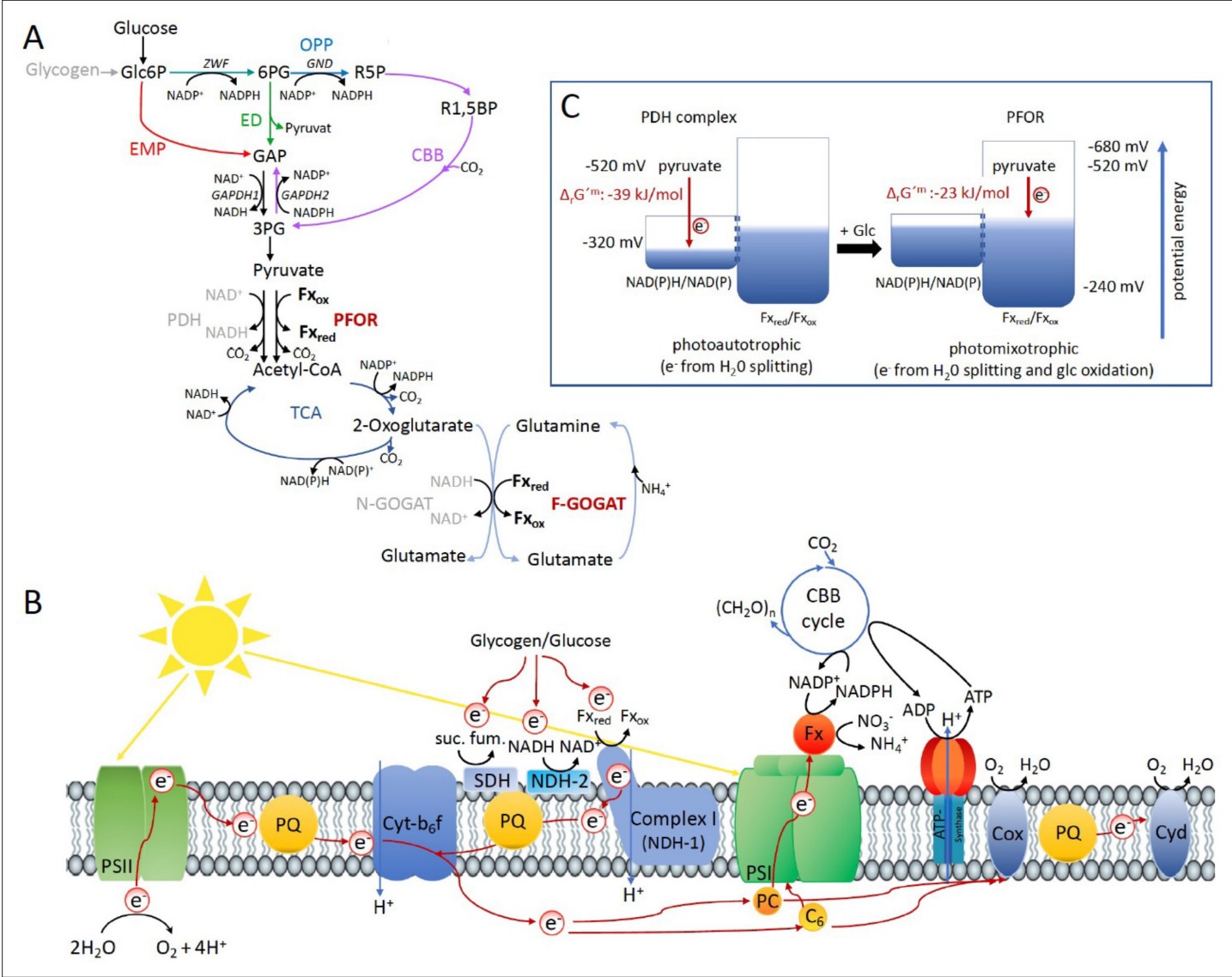

**Figure 4.** Optimal photomixotrophic growth requires low abundant ferredoxins, pyruvate:ferredoxin oxidoreductase (PFOR) and F-GOGAT. Electrons from glucose oxidation that arrive at photosystem I (PSI) require ferredoxin-dependent photosynthetic complex I (NDH-1). Cells shift from utilizing NAD(H)- to ferredoxin-dependent enzymes when brought from photoautotrophic to photomixotrophic conditions. (**A**) Glycolytic routes, lower glycolysis, and the tricarboxylic acid (TCA) cycle yield NAD(P)H from glucose oxidation. The only known enzyme that produces reduced ferredoxin from glucose oxidation is PFOR. Both the decarboxylation of pyruvate as well as the synthesis from glutamate from 2-oxoglutarate and glutamine can be catalyzed by distinct enzymes that either utilize ferredoxin (PFOR, F-GOGAT) or NAD(H) (pyruvate dehydrogenase [PDH] complex, N-GOGAT). (**B**) Photosynthetic complex I (NDH-1) accepts electrons from reduced ferredoxin. The complex is required for the input of electrons from glucose oxidation into photosynthesis in the presence of DCMU. (**C**) The $\Delta_r G'^m$ of pyruvate decarboxylation via the PDH complex is more negative that via PFOR, which results in a higher driving force (for calculations see Materials and methods). Photomixotrophy results in reducing conditions. The redox potential of the NAD(P)H/NAD(P)$^+$ pool which is around $-320$ mV will turn more negative upon reduction. This could facilitate the transfer of electrons from NADH to ferredoxins. In addition, inactivation of NAD$^+$-dependent enzymes (such as the PDH complex) and their functional replacement by ferredoxin-dependent enzymes (such as PFOR) support the suggested shift from the utilization of the NAD(H) to the ferredoxin pool.

($\Delta hk$) should no longer be able to metabolize external glucose. The glycogen phosphorylase mutant ($\Delta glgP1\Delta glgP2$) is unable to break down its internal glycogen reservoir (*Theune et al., 2021*; *Doello et al., 2018*; *Makowka et al., 2020*). As expected and in parts shown recently (*Theune et al., 2021*), addition of glucose resulted in higher donations of electrons to PSI in the WT and $\Delta glgP1\Delta glgP2$, whereas neither $\Delta ndhD1\Delta ndhD2$ nor $\Delta hk$ were able to provide electrons from glucose oxidation to PSI (*Figure 3C*). Photosynthesis using glucose oxidation and PSI thus relies on the ferredoxin-dependent photosynthetic complex I. In line with this, it was shown earlier that $\Delta ndhD1\Delta ndhD2$ is not

able to grow in the presence of glucose and DCMU under photoheterotrophic conditions (*Ohkawa et al., 2000*).

## Discussion

Under photomixotrophic conditions, photosynthesis and glucose oxidation operate in parallel. The cells are thus flooded with electrons from water oxidation at PSII and electrons from glucose oxidation (*Figure 4*). This causes highly reducing conditions in the cells as visible in reduced NAD(P)H and ferredoxin pools (*Figure 1B* and *Figure 1—figure supplement 4*). Our data indicate that the PDH complex gets inhibited at high NADH levels under these conditions and is subsequently most likely functionally replaced by PFOR (*Figures 1 and 2*). Furthermore, the cells seem to rely on a general shift from utilizing NAD(H)- to ferredoxin-dependent enzymes under these conditions. In line with this, low abundant ferredoxins, whose functions are still only partly understood in detail, and ferredoxin-dependent F-GOGAT are required for optimal photomixotrophic performance (*Figure 3*). Photosynthetic complex I (NDHI) which accepts electrons from reduced ferredoxin (*Schuller et al., 2019*), is furthermore required to feed electrons from glucose oxidation into the photosynthetic electron chain and to thereby enhance electron flow at PSI (*Figures 3 and 4*).

PFORs are with a few reported exceptions highly oxygen-sensitive enzymes that work under strictly anaerobic conditions (*Pieulle et al., 1995*; *Vita et al., 2008*; *Kerscher and Oesterhelt, 1981*). We found that PFOR of *Synechocystis* is stable in the presence of oxygen, however, in vitro we could only measure the decarboxylation of pyruvate in the absence of oxygen (*Figure 2*). However, our data strongly indicate that this enzyme is active in vivo under aerobic and highly reducing conditions.

Similar results were recently reported for *E. coli*. *E. coli* possesses three enzyme systems to convert pyruvate to acetyl CoA: the PDH complex, PFOR, and pyruvate formate-lyase (PFL) (*Blaschkowski et al., 2005*). The PDH complex and PFL are the principle enzyme systems to convert pyruvate to acetylCoA in *E. coli*, whereas PFOR is expressed at very low levels (*Blaschkowski et al., 2005*). Transcription of PFOR was shown to be enhanced under oxidative stress (*Nakayama et al., 2013*). *E. coli* decarboxylates pyruvate via the PDH complex in the presence of oxygen. Under anaerobic conditions NADH levels rise and inhibit the PDH complex. PFL gets activated and the cells switch to fermentation. PFL activation requires reduced flavodoxin which is provided by PFOR (*Blaschkowski et al., 2005*). The regulation at the pyruvate node in *E. coli* is thus mainly regulated via the availability of oxygen and its concomitant requirement for redox control and ATP (*Wang et al., 2010*). By downregulation of glucose-6P dehydrogenase (ZWF), less NADPH was produced in *E. coli*, which activated the expression of PFOR and ferredoxin reductase (FPR) (*Li et al., 2021*). PFOR and FPR were shown to contribute to stationary-phase metabolism in aerobic cultures in this mutant probably by converting reduced ferredoxin to NADPH (*Li et al., 2021*). PFOR is thus obviously involved in redox control in these mutants in the presence of oxygen. This finding was highly unexpected, as PFOR activity in crude extracts from aerobically grown *E. coli* cells is only detectable in the absence of oxygen in vitro (*Nakayama et al., 2013*). There are several reports in prokaryotes and eukaryotes on the expression of enzymes under oxic conditions that are assigned to anaerobic metabolism (*Gould et al., 2019*; *Schmitz et al., 2001*; *Gutekunst et al., 2005*). One example is the production of hydrogen by the oxygen-sensitive FeFe-hydrogenase in air-grown *Chlamydomonas reinhardtii* algae in a fully aerobic environment, which is enabled by microoxic niches within the thylakoid stroma (*Liran et al., 2016*). Another example is the constitutive expression of PFOR and the oxygen-sensitive NiFe-hydrogenase under oxic conditions in cyanobacteria. By itself, the widespread presence of these enzymes in organisms that either live predominantly aerobically as for example cyanobacteria or are even obligate aerobes as for example *S. acidocaldarius*, which possesses a PFOR, indicates a misconception and lack of understanding. The PFOR of *S. acidocaldarius* could be isolated as stable enzyme in the presence of $O_2$, however, enzyme activity measurements required the consumption of oxygen in vitro (*Witt et al., 2019*). Does this mean, that anaerobic microniches are required within this obligate aerobe to activate an enzyme of its central carbon metabolism? It might alternatively be that living cells have the ability to maintain reducing conditions in the presence of oxygen by yet unknown mechanisms that for example consume oxygen, which is a challenge in enzymatic in vitro assays. Conclusions on in vivo enzyme activities based on in vitro experiments therefore should be made with caution. Even though we could measure decarboxylation of pyruvate via PFOR only in the absence of oxygen in vitro, our data strongly indicate that this enzyme is active in vivo under aerobic and highly reducing conditions.

We assume that either anaerobic microniches or alternatively mechanisms within the cell that are not understood yet, keep the enzyme active in an aerobic environment.

Low abundant ferredoxins are required for optimal photomixotrophic growth (*Figure 3*), which is surprising when looking at glycolytic routes for glucose oxidation. Glucose is alternatively oxidized via different glycolytic routes in *Synechocystis* (*Figure 4A*). Flux analyses have shown that glycolytic intermediates enter the CBB cycle, eventually reach lower glycolysis, and finally provide pyruvate (*Nakajima et al., 2014*). Depending on the precise route taken, glucose oxidation yields distinct forms of reducing equivalents (*Makowka et al., 2020*). Three enzymes are involved in oxidation steps: Glc6P dehydrogenase (Zwf) and 6 PG dehydrogenase (Gnd) yield NADPH, whereas GAP dehydrogenase (GAPDH) yields NADH. NAD(P)H is furthermore provided downstream in the TCA cycle. PFOR is thus the only known direct source for reduced ferredoxin in glucose oxidation beside PSI (*Figure 4*). The wide network of low abundant ferredoxins in *Synechocystis* and the importance of these ferredoxins under photomixotrophic conditions on the one hand and the low number of known enzymes that directly reduce ferredoxins on the other hand unveils that our conception is not yet inherently consistent. An additional potential source of reduced ferredoxin could be the transfer of electrons from NAD(P)H. The transhydrogenase (PntAB), which is located in the thylakoid membrane utilizes proton translocation to transfer electrons from NADH to NADP$^+$ (*Kämäräinen et al., 2017*). Electrons from NADPH could be further transferred to ferredoxin via ferredoxin-NADPH-oxidoreductase (FNR). Another potential turntable for the exchange of electrons is the diaphorase part of the NiFe-hydrogenase in *Synechocystis*, which was recently shown to shuttle electrons between NAD(P)H, flavodoxin and several ferredoxins in vitro (*Artz et al., 2020*).

In order to get a complete picture of the ferredoxin network and potential interaction partners, it would be essential to know the redox potentials of all low abundant ferredoxins in *Synechocystis*. Currently, they have been determined for Fx1 (−412 mV), Fx2 (−243 mV), and Fx4 (−440 mV), whereas the value for Fx4 is based on measurements of a homologue in *Thermosynechococcus elongatus* (*Bottin and Lagoutte, 1992*; *Schorsch et al., 2018*; *Motomura et al., 2019*). Fx1–Fx6 in *Synechocystis* possess 2Fe2S clusters for which redox potentials between −240 and −440 mV are typical (*Liu et al., 2014*). For 3Fe4S clusters as present in Fx8 (containing one 3Fe4S and one 4Fe4S cluster) redox potentials between −120 and −430 mV were determined and for 4Fe4S clusters as present in Fx7 (4FeFS) and Fx9 (containing two 4Fe4S clusters) redox potentials between −300 and −680 mV were found (*Liu et al., 2014*). Our data show that Fx9 is of importance under photomixotrophic conditions (*Figure 3A*). The redox potential of Fx9 in *Synechocystis* has been assumed to be around −420 mV based on its interaction partners (*Cassier-Chauvat and Chauvat, 2014*). However, this value requires experimental validation. Without yet knowing the exact values for all ferredoxins in *Synechocystis*, it is obvious that they span a wide range of redox potentials. Our data indicate that cells perform a general shift from utilizing NAD(H)- to ferredoxin-dependent enzymes under highly reducing photomixotrophic conditions.

The following lines include theoretical reflections based on this observation. However, as these conclusions are not entirely supported by the data, they should be regarded as hypothetical and are meant as thought-provoking impulses.

The replacement of FeS enzymes and ferredoxins by FeS-free alternatives and NADPH in the course of evolution is in general discussed with regard to the oxygen sensitivity of FeS clusters in connection with the shift from anoxic to oxic conditions on Earth (*Imlay, 2006*; *Gould et al., 2019*). Oxygen is without any doubt one important factor. However, the shift from anoxic to oxic conditions went along with a shift from reducing to more oxidizing conditions. This shift was among others achieved by the escape of hydrogen into space, which irreversibly withdrew electrons from Earth (*Catling et al., 2001*). The withdrawal of electrons and the establishment of oxidizing conditions might have been an additional important factor (independent of oxygen and the oxygen sensitivity of FeS clusters) that triggered these evolutionary changes by enabling reactions with higher driving forces. The idea is thus that PFOR and ferredoxins might have been replaced by the PDH complex and NADH due to their potential to release larger amounts of Gibbs-free energy ($\Delta G < 0$). When competing with other organisms for resources an accelerated metabolism can be highly beneficial.

The decision to either utilize the PDH complex or alternatively PFOR and along this line, the replacement of PFOR by the PDH complex in the course of evolution might have been determined by the prioritization for high chemical driving forces.

On that note, we were unable to delete the PDH complex in *Synechocystis*, which points to its essential role. PFOR is in contrast dispensable under photoautotrophic conditions and cells obviously prefer to decarboxylate pyruvate via the PDH complex under these conditions. By transferring electrons to NAD$^+$ instead of ferredoxin less Gibbs-free energy is stored. However, this comes along with a higher driving force that is visible when regarding the reaction Gibbs energies of $\Delta_r G'^m$ −39 kJ/mol for the reaction catalyzed by the PDH complex versus $\Delta_r G'^m$ −23 kJ/mol for the reaction catalyzed by PFOR (*Figure 4C*; *Noor et al., 2013*).

The idea is thus that the NADH/NAD$^+$ pool gets reduced first prioritizing high driving forces. However, as the redox potential of the NADH/NAD$^+$ pool turns slowly more negative, it might reach levels that are characteristic for ferredoxin couples. This might provoke a metabolic shift to transfer electrons to oxidized ferredoxin instead of NAD$^+$ which should come along with lower metabolic rates (*Figure 4C*). This idea fits well with the observation, that PFOR and low abundant ferredoxins gain importance in the stationary growth phase (*Figures 1A and 3A*), which is characterized by a slowing down of metabolic reactions. The shift can be regulated on several levels. Among others, as shown in this study, high NADH/NAD$^+$ ratios can inactivate enzymes that rely on this couple and thereby support the action of isoenzymes that interact with the Fx$_{red}$/Fx$_{ox}$ couple instead. In addition, a shift to more reducing conditions (*Figure 1—figure supplement 4*), will alter the thermodynamic driving force of many redox reactions, and may in itself necessitate a shift in pathways. In addition, electron turntables as the transhydrogenase, FNR, and the diaphorase can support this shift (*Artz et al., 2020*; *Kämäräinen et al., 2017*).

By shifting their pools of reducing equivalents, cells are thus able to finetune their metabolism. They either liberate or save Gibbs-free energy and thereby either accelerate or slow down metabolic reactions, as required.

## Conclusion

The cyanobacterium *Synechocystis* encounters highly reducing conditions under photomixotrophy in the presence of oxygen. The PDH complex gets inactivated under these conditions at high NADH/NAD$^+$ ratios and is functionally most likely replaced by PFOR. PFOR is stable in the presence of oxygen in vitro and reduces ferredoxin instead of NAD$^+$. PFOR, low abundant ferredoxins, and the ferredoxin-dependent GOGAT are required for optimal photomixotrophic growth and performance. Electrons from the oxidation of external glucose furthermore rely upon the presence of photosynthetic complex I (which accepts electrons from ferredoxin) in order to reach PSI. These findings indicate that cells perform a general shift in the utilization of their reducing equivalent pools from NAD(H) to ferredoxin under photomixotrophic conditions.

## Materials and methods

### Bioinformatic analysis concerning the distribution of PFOR and PDH complex in cyanobacteria

All completely sequenced cyanobacterial genomes were analyzed via tblastn for the presence of the PDH complex and PFOR. For this, in order to exclude symbionts, cyanobacterial genomes were in a first step searched for the *psbD* gene (PSII subunit). We used the *psbD* gene (*sll0849*) of *Synechcoystis* as bait. Only genomes containing *psaD* were used for all further analysis. One hundred and ninety-seven genomes remained and were searched by tblastn using the *pdhA* subunit (*slr1934*) from the PDH complex from *Synechcoystis* as bait. The largest expect value was $2 \times 10^{-136}$. *pdhA* was found in all genomes analyzed. Sixty-seven of these genomes contain *nifD* (highest *e*-value $4 \times 10^{-104}$) and *nifK* (highest *e*-value $1 \times 10^{-73}$), the two subunits of the nitrogenase for N$_2$ fixation and a diazotrophic lifestyle. Diazotrophic and nondiazotrophic cyanobacteria were searched for the presence of PFOR by using *sll0741* from *Synechcoystis*. The highest *e*-value in this case was 0.

### Growth conditions

All strains were grown in 50 ml BG-11 (*Stanier et al., 1979*) buffered with TES pH 8. WT, Δ*pfor*, Δ*f-gogat*, Δ*n-gogat*, Δ*isiB*, all ferredoxin deletion mutants, Δ*ndhD1*Δ*ndhD2*, Δ*hk*, and Δ*glgP1*Δ*glgP2* were and placed in 100 ml Erlenmeyer flasks on a rotary shaker at 28°C, 50 µE/m$^2$/s and 100 rpm. After several days of growth, the cells were inoculated into 200 ml BG-11 at an OD$_{750}$ of 0.05 and placed

into glass tubes bubbled with air at 50 µE/m$^2$/s at 28°C and growth was monitored by measuring the optical density at 750 nm. In liquid cultures, all the strains were grown without addition of antibiotics and for photomixotrophic conditions 10 mM glucose was added.

For mutant selection and seggregation, the cells were grown on BG-11-agar containing 50 µg/ml kanamycin, 20 µg/ml spectinomycin, 25 µg/ml erythromycin, 10 µg/ml gentamycin, and 20 µg/ml chloramphenicol.

## Construction of mutants

All the primers used in this study are listed in *Supplementary file 1a*. All mutants are listed in *Supplementary file 1a*. All mutants were constructed in the nonmotile GT WT of *Synechocystis* sp. PCC 6803 (*Trautmann et al., 2012*). The procedure to generate the constructs for the deletion of *pfor*, *pdhA*, *isiB*, and the different ferredoxin genes was described in *Hoffmann et al., 2006*. In brief, the up- and downstream regions as well as the required antibiotic resistance cassette were amplified by PCR. Subsequently, the three fragments were combined by a PCR fusion including the outermost primers. The resulting product was inserted by TA cloning into the pCR2.1 TOPO vector (Thermo Fisher, Waltham, MA, USA). Constructs for the deletion of the genes of the NADH- and ferredoxin-dependent GOGAT were generated by Gibson cloning (*Gibson et al., 2009*) assembling three fragments into the pBluescript SK(+) in a single step. After examination by sequencing the plasmids were transformed into *Synechocystis* sp. PCC 6803 cells as described (*Williams, 1988*). Resulting transformants were either checked by PCR or Southern hybridization after several rounds of segregation (*Figure 2—figure supplement 2* and *Figure 3—figure supplement 1*). To generate a construct for overexpression of *pfor* (*sll0741*) including a His-tag a DNA fragment containing 212 bp up- and 212 bp downstream of the *sll0741* start codon, with a BamHI, XhoI, and NdeI site in between and 20-bp sequences that overlap with the pBluescript SK(+) vector at the respective ends was synthesized by GeneScript (Psicataway Township, NJ, USA). Another DNA fragment containing a modified petE promotor, followed by His-tag, TEV cleavage recognition site and linker encoding sequences, various restriction sites and 20-bp sequences that overlap with the pBluescript SK(+) vector at the respective ends was also synthesized by GenScript. These fragments were cloned into the pBluescript SK(+) vector by Gibson cloning, respectively. A kanamycin antibiotic resistance cassette was inserted into the EcoRV site of the plasmid containing the modified petE promotor. The resulting promoter-cassette plasmid and the PFOR plasmid were digested with BamHI and NdeI and the promoter cassette was ligated into the alkaline phosphatase treated PFOR plasmid to yield the final construct. This plasmid was sequenced, transformed into *Synechocystsis* sp. PCC 6803 and segregation was confirmed by PCR analysis (*Figure 3—figure supplement 1*).

## Southern blotting

200 ng genomic DNA was digested with HindIII and loaded on a 0.8% agarose gel in TBE buffer. After blotting the DNA on a nylon membrane (Hybond N+, Merck, Darmstadt, Germany) it was cross-linked to the membrane in a Stratalinker (Stratagene, CA, USA). Detection of the respective bands was carried out by the Dig DNA labeling and detection kit (Roche, Penzberg, Germany) according to the manufacturer's instructions.

## RT-PCR

To a volume of 15 µl containing 1 µg of RNA 2 µl RNase-free DNase (10 U/µl, MBI Fermentas, St. Leon-Rot, Germany), 2 µl 10× DNase buffer (MBI Fermentas, St. Leon-Rot, Germany) and 1 µl Riboblock RNase Inhibitor (40 U/µl, MBI Fermentas, St. Leon-Rot, Germany) were added before incubation at 37°C for 2 hr. Subsequently the sample was quickly cooled on ice. 2 µl 50 mM EDTA was added and it was incubated at 65°C for 10 min and again quickly cooled on ice to get rid of the DNase activity. To check the digestion efficiency, 1 µl of the sample was used as a template for PCR. 1 µl genomic DNA and 1 µl H$_2$O were used as positive and negative controls, respectively. Reverse transcription PCR was performed with 9 µl of those samples free of DNA with the RT-PCR kit (Applied Biosystems, Karlsruhe, Germany) according to the manufacturer's instruction. 9 µl of the same sample was used in parallel as a negative control. The reaction mixture was incubated for 1 hr at 37°C including a gene-specific tag-1 primer. For the subsequent PCR, a gene-specific tag-2 primer and the respective reverse primer (see *Supplementary file 1a*) were used.

## Oxygen measurements

To measure the concentration of dissolved oxygen in the cultures, oxygen sensors from Unisense (Unisense, Aarhus, Denmark) were used. After a two-point calibration of the sensor by using distilled water equilibrated with air and a solution with 0.1 M NaOH and 0.1 M ascorbate containing no oxygen it was placed in the respective culture and the measurement was started.

## Determination of NAD$^+$/NADH

All the cultures used for NAD$^+$/NADH determination experiment were grown autotrophically and mixotrophically in 250 ml BG-11 medium. 5 ml to 10 ml cells, equivalent to about $10^9$ cells/ml (10 ml cultures of OD$_{750}$ of 1) were sampled for the measurements. The cells were centrifuged at 3500 × $g$ −9°C for 10 min and the pellets were washed with 1 ml 20 mM cold phosphate-buffered saline (20 mM KH$_2$PO$_4$, 20 mM K$_2$HPO$_4$, and 150 mM NaCl). The suspension was transferred to a 2 ml reaction cup and was centrifuged at 12,000 × $g$ for 1 min at −9°C. For all further steps the NAD$^+$/NADH Quantification Colorimetric Kit (Biovision, CA, USA) was used. The pellet was resuspended in 50 µl extraction buffer and precooled glass beads (∅ = 0.18 mm) were added to about 1 mm to the surface of the liquid. The mixture was vortexed four times 1 min in the cold room (4°C) and intermittently chilled on the ice for 1 min. 150 µl extraction buffer was added again and the mixture was centrifuged at 3500 × $g$ for 10 min at −9°C. The liquid phase was transferred as much as possible into a new reaction cup and centrifuged at maximum speed for 30 min at −9°C. All further steps were conducted as described by the manufacturer. Finally, the samples were incubated for 1–4 hr in 96-well plates before measuring absorbance at 450 nm by TECAN GENios (TECAN Group Ltd, Austria) along with a NADH standard curve.

## Determination of the redox level of ferredoxin

To compare the redox level of the ferredoxin pool of autotrophic and mixotrophic WT cells the Dual-KLAS/NIR was used (*Klughammer and Schreiber, 2016*). Cells were grown for 3 days under either conditions, harvested and adjusted to 20 µg chlorophyll/ml for the measurements. The cell suspension was consecutively illuminated with increasing light intensities between 35 and 162 µE/m$^2$/s. At lower light intensities the signal was notoriously noisy and not used further. Each new light intensity was applied to the cells for 1 min to reach steady state before data acquisition started. To this end a multiple turnover pulse of 800 ms and 19,800 µE/m$^2$/s was applied six times every 24 s on top of the actinic light intensity to fully oxidize or reduce the respective component. The averaged data recorded just before, during, and after the pulse was used to determine the signal height for all three redox partners (P700, plastocyanin, and ferredoxin). This signal was divided by the maximal signal recorded by a NirMax measurement done with the same sample as described before (*Theune et al., 2021*). Under steady state conditions the FeS signal detected by the Dual-KLAS/NIR should be close to the redox state of ferredoxin since the FeS clusters of PSI should be in equilibrium with those in ferredoxin.

## Determination of the redox level of NAD(P)H

To compare the redox level of NAD(P)H of autotrophic and mixotrophic WT cells the Dual-KLAS/NIR was connected to the NADPH module (*Klughammer and Schreiber, 2016*). Cells from 3-day-old cultures were harvested and adjusted to 10 µg chlorophyll/ml for measurement. The cell suspension was consecutively illuminated with increasing light intensities between 16 and 162 µE/m$^2$/s. Each new light intensity was applied to the cells for 1 min to reach steady state before data acquisition started. In this case, a 2-s pulse of 740 µE/m$^2$/s was applied 10 times every 13 s on top of the actinic light intensity to fully reduce NAD(P)H. The data were recorded from about 4 s before the pulse to 4 s after the pulse with an average over 50 data points. After averaging all 10 measurements the signal height was determined to get an estimate on how much NAD(P)H could still be reduced. In parallel to these measurements, the oxygen evolution was measured by an oxygen microelectrode (Unisense, Aarhus, Denmark) to determine the amount of electrons available for NADP$^+$ reduction due to linear electron transport.

## Purification and activity measurement of dihydrolipoyl dehydrogenase (E3 subunit, SynLPD)

The recombinant His-tagged SynLPD (Slr1096) was generated and purified essentially as described previously (*Reinholdt et al., 2019*). Prior activity measurements, the elution fractions were desalted

through PD10 columns (GE Healthcare, Solingen, Germany). The protein concentration was determined according to *Bradford, 1976*. SynLPD activity was determined in the forward direction. DL-dihydrolipoic acid served as the substrate at a final concentration of 3 mM. SynLPD activity was followed as reduction of $NAD^+$ (included in varying concentrations, 0.1, 0.2, 0.3, 0.4, 0.5, 1, 2, 3, 4, and 5 mM) at 340 nm. The $K_i$ constant was estimated in the presence of four NADH concentrations (0, 0.1, 0.15, and 0.2 mM) as well as NADPH (0.1 mM) as control. Specific enzyme activity is expressed in µmol NADH per $min^{-1}$ mg $protein^{-1}$ at 25°C. Mean values and standard deviations were calculated from at least three technical replicates for all substrate/cosubstrate combinations. All chemicals were purchased from Merck (Darmstadt, Germany).

## Purification of PFOR

For the purification of PFOR from *Synechocystis* sp. PCC 6803, three 6 l autotrophic cultures of the PFOR overexpression strain (PFOR:oe) were grown to an $OD_{750}$ of about 1. Cells were harvested by centrifugation at 4000 rpm in a JLA-8.1000 rotor for 20 min at 4°C. Initially, His-PFOR overexpression in the 6 l cultures was assessed by Sodium dodecyl sulfate–polyacrylamide gel electrophoresis analysis followed by immunoblotting with a His-tag-specific antibody (GenScript; *Figure 3—figure supplement 1*). A specific band could be detected in the overexpression mutant, confirming expression and stable accumulation of the overexpressed and N terminally His-tagged PFOR protein. For large-scale purification cells were resuspended in lysis buffer (50 mM $NaPO_4$, pH = 7.0; 250 mM NaCl; one tablet complete protease inhibitor EDTA free (Roche, Basel, Switzerland) per 50 ml) and broken by passing them through a French Press cell at 1250 p.s.i. twice. Unbroken cells and membranes were pelleted in a Beckman ultracentrifuge using a 70 Ti rotor at 35,000 rpm for 45 min at 4°C. The decanted soluble extract was adjusted to a volume of 90 ml with lysis buffer and incubated with 10 ml TALON cobalt resin (Takara, Shiga, Japan) for 1 hr at 4°C. The resin was then washed extensively with 200 ml lysis buffer and subsequently with 100 ml lysis buffer containing 5 mM imidazole. Bound proteins were eluted with 20 ml elution buffer (50 mM $NaPO_4$, pH = 7.0; 250 mM NaCl; 500 mM imidazole). The protein was concentrated overnight to a volume of 2 mL in a Vivaspin 20 Ultrafiltration Unit (5 kDa MWCO)(Merck, Darmstadt, Germany) and then loaded onto a HiLoad 26/60 Superdex TM 75 prep grade (GE Healthcare, Chicago, IL, USA) using 25 mM $NaPO_4$, pH=7.0; 50 mM NaCl; 5% (v/v) glycerol as the running buffer. The run was monitored at 280 nm and fractions were collected (*Figure 3—figure supplement 2A*).

## Acitivity measurement of PFOR

The specific activity of the PFOR was measured essentially as described (*Witt et al., 2019*). The activity assay contained in 1 ml 100 mM Tris–HCl (pH 8), 0.5 mM Coenzyme A, 10 mM pyruvate, 5 mM thiamine pyrophosphate, 40 mM glucose, 40 U glucose oxidase, 50 U catalase, and 10 mM methyl viologen. The reduction of methylviologen was followed at 604 nm and an extinction coefficient of 13.6 mM/cm was used. The reaction was started by adding $8.9 \times 10^{-5}$ M isolated PFOR.

We also tested ferredoxin reduction by the PFOR by a mixture containing the same substances as above except methyl viologen. To this mixture 1.6 mM ferredoxin 1 and 1.3 mM ferredoxin:$NADP^+$ reductase and 1 mM $NADP^+$ were added. In this case, the reduction of $NADP^+$ was followed at 340 nm. The same mixture without glucose, glucose oxidase, and catalase were used to test if the enzyme also works in the presence of oxygen.

## In vivo electron flow through PSI

The electron flux through PSI was measured by the Dual-KLAS/NIR (Walz GmbH, Effeltrich, Germany) by a newly developed method (*Theune et al., 2021*). In brief, cell suspensions were adjusted to 20 µg/ml chlorophyll and 20 µM DCMU was added. Electron flow through PSI was determined by dark-interval relaxation kinetics (DIRK) measurements at a light intensity of 168 µE/$m^2$/s in the absence and presence of 10 mM glucose.

## Determination of reaction Gibbs energies

$\Delta_r G'^m$ for the reaction catalyzed by the PDH complex and by PFOR was calculated using eQuilibrator (http://eauilibrator.weizmann.ac.il/) according to *Noor et al., 2013*. $CO_2$ (total) was considered as hydrated and dehydrated forms of $CO_2$ are considered to be in equilibrium in biochemical reactions.

Ionic strength of 0.2 M, pH of 7, and metabolite concentrations of 1 mM were assumed. In order to determine the redox potential of pyruvate we used the reactions Gibbs energy of −39 kJ/mol for the PDH complex and −23 kJ/mol for PFOR. Assuming a redox potential of −320 mV for NAD(P)H and −400 mV for ferredoxin the potential of pyruvate was determined according to $\Delta G = -nF\Delta E$ to −520 mV.

## Acknowledgements

This study was supported by grants from the China Scholarship Council (CSC) (Grant # 201406320187), FAZIT-Stiftung, Deutsche Bundesstiftung Umwelt, German Ministry of Science and Education (BMBF FP309), and the German Science Foundation (DFG Gu1522/2-1, HA2002/23-1, and FOR2816).

## Additional information

### Funding

| Funder | Grant reference number | Author |
|---|---|---|
| Deutsche Forschungsgemeinschaft | DFG Gu1522/2-1 | Kirstin Gutekunst |
| China Scholarship Council | 201406320187 | Yingying Wang |
| Deutsche Forschungsgemeinschaft | HA2002/23-1 | Thomas Barske |
| Deutsche Forschungsgemeinschaft | FOR2816 | Martin Hagemann Kirstin Gutekunst |
| Bundesministerium für Bildung und Forschung | BMBF FP309 | Marko Boehm Jens Appel |

The funders had no role in study design, data collection, and interpretation, or the decision to submit the work for publication.

### Author contributions

Yingying Wang, Conceptualization, Data curation, Investigation, Methodology, Writing - original draft, Writing - review and editing; Xi Chen, Katharina Spengler, Karoline Terberger, Thomas Barske, Stefan Timm, Data curation, Investigation, Methodology, Writing - review and editing; Marko Boehm, Jens Appel, Data curation, Investigation, Methodology, Supervision, Writing - review and editing; Natalia Battchikova, Conceptualization, Data curation, Investigation, Writing - review and editing; Martin Hagemann, Conceptualization, Formal analysis, Funding acquisition, Supervision, Writing - review and editing; Kirstin Gutekunst, Conceptualization, Formal analysis, Funding acquisition, Project administration, Supervision, Writing - original draft, Writing - review and editing

### Author ORCIDs

Yingying Wang http://orcid.org/0000-0002-0603-6691
Stefan Timm http://orcid.org/0000-0003-3105-6296
Kirstin Gutekunst http://orcid.org/0000-0003-4366-423X

### Decision letter and Author response

Decision letter https://doi.org/10.7554/eLife.71339.sa1
Author response https://doi.org/10.7554/eLife.71339.sa2

## Additional files

### Supplementary files

• Supplementary file 1. List of primers. (a) List of primers used in this study to generate deletion strains and for RT-PCR. (b) List of *Synechocystis* strains and mutants used in this study.

• Transparent reporting form

## Data availability

All data generated or analysed during this study are included in the manuscript and supporting file; Source data files have been provided for Figures 1, 2, 3 and accompanying figure supplements.

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
