## [Editor Report]

In this study, the authors detail evidence supporting a role of pyruvate:ferredoxin oxidoreductase under oxygenic conditions. The works provides explanations for why "anaerobic enzymes" can be present and advantageous under aerobic conditions.

---

## [Decision Letter]

**Decision letter after peer review:**

Thank you for submitting your article "Pyruvate:ferredoxin oxidoreductase and low abundant ferredoxins support aerobic photomixotrophic growth in cyanobacteria" for consideration by *eLife*. Your article has been reviewed by 3 peer reviewers, and the evaluation has been overseen by a Reviewing Editor and Gisela Storz as the Senior Editor. The following individuals involved in review of your submission have agreed to reveal their identity: Daniel C Ducat (Reviewer #1); Robert Burnap (Reviewer #2); Wolfgang Nitschke (Reviewer #3).

The work demonstrates that PFOR-mediated decarboxylation of pyruvate replaces the (more canonical) PDH-based pathway cells of the cyanobacterium Synechocystis sp. PCC6803 are shifted from photoautotrophic to photomixotrophic growth conditions. The reviewers recognized the novelty and potential importance of the work, in particular on the implications for the evolution of different enzymes and metabolic pathways under aerobic and anaerobic conditions.

1. On the other hand, all of the reviewers had similar criticisms that the text was greatly over-interpreted the results, while ignoring some possible alternative interpretations, and needs to be extensively rewritten. There needs to be a complete overhaul of both the Introduction and Discussion to address the points made by the reviewers.

2. Multiple reviewers specifically cited problems with the data in Figure S4, which appeared unconvincing or even contradictory to the stated interpretation. At the very least, this issue needs to be addressed in the text, if not in the data itself.

3. Most of the results are based on growth data, but was interpretated in the context of cellular redox states. It was disappointing that the efforts to probe cellular redox states were not very strong, as described by Reviewer 1, weakening the arguments (see below).

4. The consensus of the reviewers was that such measurements would greatly strengthen the paper. Alternatively, it may be possible to modify the text to reflect the alternative models and potential ambiguities inherent in the approach.

5. A major component of the proposed model is that PFOR is specifically active highly aerobic conditions, but this claim is never directly tested, and evidence from simple deletion mutant without tested for redox states activity does not fully test this possibility, see comments by Reviewer 2.

6. Further, the data was interpreted in the context of redox state, but as Reviewer 3 points out, the available thermodynamic driving force should be very different under the different growth conditions, and may in itself necessitate a shift in pathway, Thus, the results can be reasonably interpreted without the need to involve "inactivation" of enzymes. This, or course, could be settled by enzymatic assays and redox probes.

*Reviewer #2 (Recommendations for the authors):*

Most of my comments are already present in the public review. However, the authors should seriously consider making measurements of cells in vivo where they're tracking pyridine nucleotide (blue-green) fluorescence. Such measurements are primarily tracking and NADPH rather than NADH, nevertheless it could prove very informative regarding the ability for the cells to assimilate photosynthetic reductant as the light comes on. Such measurements can be performed using a PAM fluorometer, for example, with the appropriate attachments. Also, contemporaneous measurements of chlorophyll fluorescence can provide valuable information on linear and cyclic photosynthetic electron flow.

*Reviewer #3 (Recommendations for the authors):*

I very much like the performed experiments and find that they are well-planned, professionally performed and that they provide solid evidence for the discussed switch of pathways. However, I feel that the discussion is unconvincing in certain places and tries to convey concepts which I would consider doubtful at the least. I will be happy if the authors can prove me wrong.

– Overly affirmative preconception: on line 46 and also subsequently in the manuscript, you claim that ferredoxins are older than NAD. You base your statement on the observation that ferredoxins are present in all three domains (I think that we have moved on from Woose's royalist kingdoms to the term "domains";-) of life. Isn't this also true for NAD? You may well be right that FeS-clusters preceeded NAD during life's emergence but so far I don't think that we have any evidence for LUCA lacking NAD. In your scenario NAD would even come in only much later when the environment became more oxidizing.

Reducing and oxidizing conditions:

– eg. l 289 ll: it is difficult for me to see how you imagine maintaining reducing conditions in the presence of oxygen (in any setting, cells or in vitro or whatever). Of course, it all depends on what you call reducing … However, in your manuscript, reducing is always connoted with NADH or ferredoxin, so well below the O2/superoxide transition. If you have O2 under such "reducing" conditions and some undefined one-electron compound (e.g. ferredoxins) floating around, they have no choice but reduce O2 to O2.-. It then all depends on who is dominant (in abundancy), the reductant or O2. If it is O2, then you will loose your reducing conditions, if it is the reductant, you will turn anaerobic. So I honestly don't understand what you mean by "reducing conditions in the presence of oxygen". By contrast, spatial separation (your first alternative) is of course an option.

– l303 ll: You claim that loss of hydrogen to space will "withdraw electrons and …[thereby] … establish oxidizing conditions". You have reducing conditions if there is an abundance of reductant and oxidizing conditions if you have an abundance of oxidant. Reducing the amount of reductant therefore doesn't take you to oxidizing conditions! On an O2-free planet, the reductants may have been H2, Fe2+, CH4, formate etc and the oxidants CO2, SO42-, Fe3+ and the likes. On line 305 you say that the loss of H2 would lead to higher driving forces. How? You don't add any new (more positive) oxidant, you only take out reductant. Therefore the driving force for any individual electron transfer reaction will stay put while the deltaG of the bulk reaction will even decrease (since you now have lower amounts of reductant).

– l366 ll: I can see what you want to say on these lines and you are basically correct. However, this is how you would explain the situation to a colleague from the humanities (no condescension intended!). I would hold that in a life science article you can assume that the reader is able to digest the Nernst equation and therefore comprehend that there is a mathematical relationship between the standard (or midpoint) potential at equal amount of oxidized and reduced form of a redox couple and the effective potential at given ratios of ox to red.

– 370 ll: "slow down the back reaction and speed up the forward reaction". Ambiguous and potentially wrong. The reaction ks are of course unaffected but mass-action results in shifted equilibrium concentrations of donor and acceptor when deltaE changes.

– 372 ll: (my main concern regarding the entire manuscript!) You go on the conclude that upon increasing NADH over NAD+ you shift the effective potential more negative. Perfectly correct! Now here is my question: On lines 183 – 198 of the Results section, you hypothesize on an inactivation of PDH via binding of NAD to the E3 subunit. Isn't it enough that you have increasing levels of NADH? (they call this product inhibition) To my mind, the switch from PDH (using NAD+) to PFOR (using Fd) is a thermodynamic necessity since PDH would cease to operate at very high NADH/NAD+ ratios anyway. Oxygenic photosynthesis in general has a problem of redox equilibration: There is an essentially infinite pool of electron donor (water) and a very limited pool of electron acceptor (CO2 to generate biomass). Hence all the regulation and cyclic transfer and overflow valves and what have you. Of course the situation is aggravated when the beasts go photomixotrophic since they are flooded with viciously reducing electrons (without having seen a single photon). Their intracellular redox poise will rapidly go to red alert reducing (NB: life needs electron transfer from red to ox to make ATP on the way; when all gets reduced through, doom looms!). Which brings me to my final question: Have you checked that your photomixotrophic cultures don't produced H2 by any chance? Channeling electrons through PFOR into the Fd pool would potentially open up the way to hydrogenase and thereby help getting rid of excess reducing equivalents via the emergency valve "proton reduction".

---

## [Author Response]

The work demonstrates that PFOR-mediated decarboxylation of pyruvate replaces the (more canonical) PDH-based pathway cells of the cyanobacterium Synechocystis sp. PCC6803 are shifted from photoautotrophic to photomixotrophic growth conditions. The reviewers recognized the novelty and potential importance of the work, in particular on the implications for the evolution of different enzymes and metabolic pathways under aerobic and anaerobic conditions.1. On the other hand, all of the reviewers had similar criticisms that the text was greatly over-interpreted the results, while ignoring some possible alternative interpretations, and needs to be extensively rewritten. There needs to be a complete overhaul of both the Introduction and Discussion to address the points made by the reviewers.

We thank all reviewers for reading our manuscript thoroughly and for raising very interesting discussion points. We addressed all raised points either by responding to the reviewers directly and by overhauling both the introduction and the discussion part.

The introduction was extended by alternative interpretations. Furthermore, animated by the requests of the reviewers we searched the literature again and found a recent study from *E. coli*, which was added both in the introduction and discussion. The authors found that PFOR in *E. coli* supports metabolism during stationary phase in the presence of oxygen in an E coli. mutant (1).

The discussion was completely reorganized and overhauled and now starts with a summary of the findings that are best supported by our data. We then added two sentences to stress that the following lines include hypothetical consideration, as suggested:

“The following lines include theoretical reflections based on the presented data. However, as these conclusions are not entirely supported by the data, they should be regarded as hypothetical and are meant as thought-provoking impulses.”

We very much hope that this overhaul is satisfying.

2. Multiple reviewers specifically cited problems with the data in Figure S4, which appeared unconvincing or even contradictory to the stated interpretation. At the very least, this issue needs to be addressed in the text, if not in the data itself.

We agree that data from Figure S4 (kinase mutants) as well as data concerning the phosphorylation of the PDH complex are not convincing. We were uncertain from the very beginning if it would be a good idea to include these data sets and having this in mind discussed the data very cautiously. However, these data are not really required for the statement of our manuscript, as the enzymatic tests on the E3 subunit of the PDH complex at different NADH concentration convincingly show, that the PDH complex gets inhibited at high NADH levels. Therefore, we now decided to take these data sets (kinase mutants Figure S4 and phosphorylation of the PDH complex) out of the manuscript. We thank the reviewers for their criticism as we feel that the manuscript improved and is more focused now.

3. Most of the results are based on growth data, but was interpretated in the context of cellular redox states. It was disappointing that the efforts to probe cellular redox states were not very strong, as described by Reviewer 1, weakening the arguments (see below).

This is an important point. Thank you very much. We agree with this criticism and now addressed it by including in vivo measurements of the redox state of NAD(P) and ferredoxin. The presented results strongly support our original interpretation that the redox state is more reduced under photomixotrophic conditions compared to photoautotrophic conditions (see new Figure 1 —figure supplement 4).

4. The consensus of the reviewers was that such measurements would greatly strengthen the paper. Alternatively, it may be possible to modify the text to reflect the alternative models and potential ambiguities inherent in the approach.

The new results were added in Figure 1 —figure supplement 4 and the text was modified accordingly, stating that both NAD(P)H and ferredoxin pools are reduced to a greater extent under photomixotrophic conditions.

5. A major component of the proposed model is that PFOR is specifically active highly aerobic conditions, but this claim is never directly tested, and evidence from simple deletion mutant without tested for redox states activity does not fully test this possibility, see comments by Reviewer 2.

We absolutely agree that direct testing of the PFOR under different conditions and oxygen concentrations would be much more convincing. Such tests have been unsuccessful in our hands for the cyanobacterial enzyme until now. It can be very difficult to detect activity in the presence of oxygen in vitro for redox enzymes, as there might be mechanisms inside the cell that continuously reactivate the enzyme upon inactivation. This is e.g. known for FeFe-hydrogenases that are irreversibly inactivated by oxygen in vitro but stay active or can be reactivated in vivo. The mimic of such reactivation was not accomplished in vitro, yet. Therefore, for the time being we are unfortunately unable to provide further evidence. However, we think that our collected data strongly indicates that PFOR is an important enzyme also under aerobic conditions and discuss the presented evidence with appropriate caution. In addition, animated by suggestions of the reviewers we found a recent report on an *E. coli* mutant, in which PFOR contributes to the metabolic stationary phase of aerobic cultures even though the enzyme requires anaerobic conditions in the in vitro test (1, 2). This report is now mentioned in the introduction and discussion.

6. Further, the data was interpreted in the context of redox state, but as Reviewer 3 points out, the available thermodynamic driving force should be very different under the different growth conditions, and may in itself necessitate a shift in pathway, Thus, the results can be reasonably interpreted without the need to involve "inactivation" of enzymes. This, or course, could be settled by enzymatic assays and redox probes.

We agree with the point raised by Reviewer 3. Our data show that high NADH levels inactivate the PDH complex. However, this does not rule out that the general shift to more reducing conditions as shown by our new measurements (Figure 1 —figure supplement 4), which will change the thermodynamic driving force of redox reactions, in itself necessitates shifts in pathways and might be e.g. valid for the shift between the importance of N-GOGAT relative to F-GOGAT. Both mechanisms might be of importance. This point was added in line 363 of the discussion.

“In addition, a shift to more reducing conditions (Figure 1 —figure supplement 4), will alter the thermodynamic driving force of many redox reactions, and may in itself necessitate a shift in pathways.”

Reviewer #2 (Recommendations for the authors):Most of my comments are already present in the public review. However, the authors should seriously consider making measurements of cells in vivo where they're tracking pyridine nucleotide (blue-green) fluorescence. Such measurements are primarily tracking and NADPH rather than NADH, nevertheless it could prove very informative regarding the ability for the cells to assimilate photosynthetic reductant as the light comes on. Such measurements can be performed using a PAM fluorometer, for example, with the appropriate attachments. Also, contemporaneous measurements of chlorophyll fluorescence can provide valuable information on linear and cyclic photosynthetic electron flow.

Thank you very much for this comment. As already stated, the NAD(P)H-module of the PAM does not distinguish NADPH and NADH. However, using this device we now show that NAD(P)H is more strongly reduced under photomixotrophy. In addition, with the Dual-KLAS/NIR we also have evidence that this is also true for the ferredoxins (newly added results in Figure 1 —figure supplement 4).

Reviewer #3 (Recommendations for the authors):As stated in the public review parts, I very much like the performed experiments and find that they are well-planned, professionally performed and that they provide solid evidence for the discussed switch of pathways. However, I feel that the discussion is unconvincing in certain places and tries to convey concepts which I would consider doubtful at the least. I will be happy if the authors can prove me wrong.– Overly affirmative preconception: on line 46 and also subsequently in the manuscript, you claim that ferredoxins are older than NAD. You base your statement on the observation that ferredoxins are present in all three domains (I think that we have moved on from Woose's royalist kingdoms to the term "domains";-) of life. Isn't this also true for NAD? You may well be right that FeS-clusters preceeded NAD during life's emergence but so far I don't think that we have any evidence for LUCA lacking NAD. In your scenario NAD would even come in only much later when the environment became more oxidizing.

Thank you very much for this point. We did not mean to say that LUCA lacked NAD. The point we wanted to make was rather, that NAD gained importance over oxygen sensitive ferredoxins during the evolution from an anaerobic to an aerobic metabolism as also stated by Gould et al., 2019 (6). In order to be more precise, we now stressed that NAD(P)H gained importance to make sure that it was as well present already.

Accordingly, we changed the former sentence “In addition, ferredoxins have in general been complemented or replaced by NAD(P)H as alternative, oxygen-insensitive reducing agents in aerobes” in line 80 to: “In addition, NAD(P)H has gained importance as alternative, oxygen-insensitive reducing agent in aerobes and thereby complemented or replaced oxygen sensitive ferredoxins, that are useful for anaerobes.”

Reducing and oxidizing conditions:– eg. l 289 ll: it is difficult for me to see how you imagine maintaining reducing conditions in the presence of oxygen (in any setting, cells or in vitro or whatever). Of course, it all depends on what you call reducing … However, in your manuscript, reducing is always connoted with NADH or ferredoxin, so well below the O2/superoxide transition. If you have O2 under such "reducing" conditions and some undefined one-electron compound (e.g. ferredoxins) floating around, they have no choice but reduce O2 to O2.-. It then all depends on who is dominant (in abundancy), the reductant or O2. If it is O2, then you will loose your reducing conditions, if it is the reductant, you will turn anaerobic. So I honestly don't understand what you mean by "reducing conditions in the presence of oxygen". By contrast, spatial separation (your first alternative) is of course an option.

This is again a good point. Thank you very much. We agree fully. We changed the sentence by adding, that mechanisms might be present that consume the oxygen.

“It might alternatively be that living cells have the ability to maintain reducing conditions in the presence of oxygen by yet unknown mechanisms that e.g. consume oxygen, which is a challenge in enzymatic in vitro assays.” (line 287 following)

– l303 ll: You claim that loss of hydrogen to space will "withdraw electrons and …[thereby] … establish oxidizing conditions". You have reducing conditions if there is an abundance of reductant and oxidizing conditions if you have an abundance of oxidant. Reducing the amount of reductant therefore doesn't take you to oxidizing conditions! On an O2-free planet, the reductants may have been H2, Fe2+, CH4, formate etc and the oxidants CO2, SO42-, Fe3+ and the likes. On line 305 you say that the loss of H2 would lead to higher driving forces. How? You don't add any new (more positive) oxidant, you only take out reductant. Therefore the driving force for any individual electron transfer reaction will stay put while the deltaG of the bulk reaction will even decrease (since you now have lower amounts of reductant).

We indeed think that the loss of hydrogen established oxidizing conditions. The oxidant was present by water splitting at PSII and the evolution of oxygen. However, the accumulation of oxygen was only possible due to the loss of reductant (H2 into space on the one hand and burial of reduced carbon on the other hand). Without this loss of reductant, respiration would have consumed all the oxygen quickly and nothing would have changed. However, by the withdrawal of H2 and reduced carbon from the equation, oxygen could accumulate. Which means that the high abundance of oxidant (O2) indeed required the removal of reductant (H2).

– l366 ll: I can see what you want to say on these lines and you are basically correct. However, this is how you would explain the situation to a colleague from the humanities (no condescension intended!). I would hold that in a life science article you can assume that the reader is able to digest the Nernst equation and therefore comprehend that there is a mathematical relationship between the standard (or midpoint) potential at equal amount of oxidized and reduced form of a redox couple and the effective potential at given ratios of ox to red.

We deleted this paragraph as suggested, when we rewrote and overhauled the discussion.

– 370 ll: "slow down the back reaction and speed up the forward reaction". Ambiguous and potentially wrong. The reaction ks are of course unaffected but mass-action results in shifted equilibrium concentrations of donor and acceptor when deltaE changes.

We fully agree with this statement. However, we think that our text was not clear enough as there might to be a misunderstanding. The point we wanted to make is related to a shift from using ferredoxin instead of NAD^+^ as a redox acceptor. In this case reaction ks are affected and deltaE decreases. This might not have been clear enough. However, as we rewrote and completely overhauled the discussion as suggested, these sentences were deleted.

– 372 ll: (my main concern regarding the entire manuscript!) You go on the conclude that upon increasing NADH over NAD+ you shift the effective potential more negative. Perfectly correct! Now here is my question: On lines 183 – 198 of the Results section, you hypothesize on an inactivation of PDH via binding of NAD to the E3 subunit. Isn't it enough that you have increasing levels of NADH? (they call this product inhibition) To my mind, the switch from PDH (using NAD+) to PFOR (using Fd) is a thermodynamic necessity since PDH would cease to operate at very high NADH/NAD+ ratios anyway.

In *E. coli* an alteration in a single amino acid of the E3 subunit of the PDH complex reduced the sensitivity of the PDH complex to inhibition via NADH and allowed decarboxylation of pyruvate at high NADH levels (9). The respective amino acid was not part of the binding pocket for NAD^+^. The mechanisms is thus not a simple product inhibition. NADH seems to have a regulatory function here and seems to bind as a regulator at a site that does not belong to the active site of the enzyme. The enzyme activity measurements that we did with the E3 subunit of Synechocystis at different NADH concentration does not allow to determine if NADH inhibited the enzyme in a regulatory manner or in the manner of classical product inhibition. We therefore left this point open.

Oxygenic photosynthesis in general has a problem of redox equilibration: There is an essentially infinite pool of electron donor (water) and a very limited pool of electron acceptor (CO2 to generate biomass). Hence all the regulation and cyclic transfer and overflow valves and what have you. Of course the situation is aggravated when the beasts go photomixotrophic since they are flooded with viciously reducing electrons (without having seen a single photon). Their intracellular redox poise will rapidly go to red alert reducing (NB: life needs electron transfer from red to ox to make ATP on the way; when all gets reduced through, doom looms!). Which brings me to my final question: Have you checked that your photomixotrophic cultures don't produced H2 by any chance? Channeling electrons through PFOR into the Fd pool would potentially open up the way to hydrogenase and thereby help getting rid of excess reducing equivalents via the emergency valve "proton reduction".

Yes, we checked the cultures for hydrogen production, but we could not detect any. The hydrogenase is very likely not able to produce hydrogen due to the presence of oxygen. A valve to get rid of excess reducing equivalents that the cells use under photomixotrophic conditions is e.g. to enhance their glycogen levels.

References

1. S. Li *et al.*, Dynamic control over feedback regulatory mechanisms improves NADPH flux and xylitol biosynthesis in engineered *E. coli*. *Metab Eng* 64, 26-40 (2021).

2. T. Nakayama, S. Yonekura, S. Yonei, Q. M. Zhang-Akiyama, *Escherichia coli* pyruvate:flavodoxin oxidoreductase, YdbK – regulation of expression and biological roles in protection against oxidative stress. *Genes Genet Syst* 88, 175-188 (2013).

3. A. Witt, R. Pozzi, S. Diesch, O. Hädicke, H. Grammel, New light on ancient enzymes – in vitro CO2 Fixation by Pyruvate Synthase of Desulfovibrio africanus and Sulfolobus acidocaldarius. *The FEBS Journal* 286, 4494-4508 (2019).

4. C. Cassier-Chauvat, F. Chauvat, Function and Regulation of Ferredoxins in the Cyanobacterium Synechocystis PCC6803: Recent Advances. *Life* 4, 666-680 (2014).

5. M. Müller *et al.*, Biochemistry and Evolution of Anaerobic Energy Metabolism in Eukaryotes. *Microbiology and Molecular Biology Reviews* 76, 444 (2012).

6. S. B. Gould et al., Adaptation to life on land at high O2 via transition from ferredoxin-to NADH-dependent redox balance. Proceedings of the Royal Society B: Biological Sciences 286, 20191491 (2019).

7. O. Schmitz, J. Gurke, H. Bothe, Molecular evidence for the aerobic expression of nifJ, encoding pyruvate : ferredoxin oxidoreductase, in cyanobacteria. *FEMS Microbiol. Lett.* 195, 97-102 (2001).

8. K. Gutekunst *et al.*, LexA regulates the bidirectional hydrogenase in the cyanobacterium Synechocystis sp. PCC 6803 as a transcription activator. *Molecular Microbiology* 58, 810-823 (2005).

9. Z. Sun *et al.*, Amino acid substitutions at glutamate-354 in dihydrolipoamide dehydrogenase of *Escherichia coli* lower the sensitivity of pyruvate dehydrogenase to NADH. *Microbiology* 158, 1350-1358 (2012).